# Exploring Metabolomics to Innovate Management Approaches for Fall Armyworm (*Spodoptera frugiperda* [J.E. Smith]) Infestation in Maize (*Zea mays* L.)

**DOI:** 10.3390/plants13172451

**Published:** 2024-09-02

**Authors:** Jayasaravanan Desika, Kalenahalli Yogendra, Sundararajan Juliet Hepziba, Nagesh Patne, Bindiganavile Sampath Vivek, Rajasekaran Ravikesavan, Sudha Krishnan Nair, Jagdish Jaba, Thurapmohideen Abdul Razak, Subbiah Srinivasan, Nivedita Shettigar

**Affiliations:** 1V.O.C. Agricultural College and Research Institute, Tamil Nadu Agricultural University (TNAU), Killikulam 628252, India; desikasara18@gmail.com (J.D.); abdulrazak.t@tnau.ac.in (T.A.R.); srinivasan.s@tnau.ac.in (S.S.); 2International Maize and Wheat Improvement Center (CIMMYT), Hyderabad 502324, India; p.nagesh@cgiar.org (N.P.); sudha.nair@cgiar.org (S.K.N.); niveditashettigar@gmail.com (N.S.); 3International Crops Research Institute for the Semi-Arid Tropics (ICRISAT), Hyderabad 502324, India; yogendra.kalenahalli@icrisat.org (K.Y.); jagdish.jaba@icrisat.org (J.J.); 4Centre for Plant Breeding & Genetics, Tamil Nadu Agricultural University (TNAU), Coimbatore 641003, India; ravikesavan@tnau.ac.in; 5Department of Genetics and Plant Breeding, Professor Jayashankar Telangana State Agricultural University (PJTSAU), Hyderabad 500030, India

**Keywords:** metabolomics, fall armyworm, maize, resistance, biotic stress

## Abstract

The Fall armyworm (FAW), *Spodoptera frugiperda* (J. E. Smith), is a highly destructive lepidopteran pest known for its extensive feeding on maize (*Zea mays* L.) and other crops, resulting in a substantial reduction in crop yields. Understanding the metabolic response of maize to FAW infestation is essential for effective pest management and crop protection. Metabolomics, a powerful analytical tool, provides insights into the dynamic changes in maize’s metabolic profile in response to FAW infestation. This review synthesizes recent advancements in metabolomics research focused on elucidating maize’s metabolic responses to FAW and other lepidopteran pests. It discusses the methodologies used in metabolomics studies and highlights significant findings related to the identification of specific metabolites involved in FAW defense mechanisms. Additionally, it explores the roles of various metabolites, including phytohormones, secondary metabolites, and signaling molecules, in mediating plant–FAW interactions. The review also examines potential applications of metabolomics data in developing innovative strategies for integrated pest management and breeding maize cultivars resistant to FAW by identifying key metabolites and associated metabolic pathways involved in plant–FAW interactions. To ensure global food security and maximize the potential of using metabolomics in enhancing maize resistance to FAW infestation, further research integrating metabolomics with other omics techniques and field studies is necessary.

## 1. Introduction

Maize (*Zea mays* L.) is the most versatile crop among cereals with respect to its adaptability, types, and uses. It is the second most widely grown crop in the world and the third most important food crop cultivated in the tropics, sub-tropics, and temperate climates, and comprises several types, such as field corn, sweetcorn, popcorn, and baby corn. It is an important crop for billions of people as food, feed, and industrial raw material. Besides serving as a staple food, maize finds extensive applications across various industries, including the production of starch, oil, protein, alcoholic beverages, sweeteners, cosmetics, and biofuel. Notably, the starch and feed sectors consume 83% of maize output, elevating its status as an industrial crop [1]. Currently, nearly 1147.7 million metric tons (MT) of maize are produced by more than 170 countries, covering an area of 203.4 million hectares (ha) with an average productivity of 5.71 t/ha [2].

However, maize productivity is significantly threatened by biotic stresses, primarily from insect pests such as Fall armyworm (FAW) and corn earworm (*Helicoverpa zea* [Boddie]) [3,4] along with other challenges such as corn rust (*Puccinia sorghi* [Schwein]) and ear rot (*Fusarium verticillioides* [Sacc.] Nirenberg) (Figure 1). Among these threats, the invasive nature and extensive damage caused by the FAW, particularly on maize, sorghum, and other millet crops, are of particular concern [5,6]. Originating in the Western Hemisphere, FAW has rapidly spread to over 109 countries across Africa, the Near East, and some south European countries and Asia, leading to substantial yield reductions, with reported losses as high as 33% in maize alone [7,8].

Efforts to combat FAW infestation have led to the development of insect-resistant maize breeding lines through conventional methods, albeit with limitations in the speed and efficacy of resistance integration into elite breeding lines [9,10]. Various control strategies have been proposed to mitigate FAW incidence, including host plant resistance, cultural methods, biological controls, biopesticides, mating disruption technologies, synthetic pesticides, and agroecological management. Despite their potential, many of these technologies face regulatory hurdles, limiting their accessibility to farmers, particularly in regions such as Asia and Africa [11,12].

In contrast, metabolomics emerges as a promising approach for FAW prevention by focusing on understanding plant metabolic profiles and their responses to insect pests [13]. Targeted metabolomics aim to elucidate the roles of specific metabolites in defense mechanisms by quantifying their levels during stress responses, while non-targeted metabolomics provide comprehensive metabolic profiles, facilitating the discovery of defense mechanisms, biomarkers, and signaling molecules. The difference between targeted and non-targeted metabolomics is formulated in Table 1. Studying metabolomic pathways in plant–pathogen/pest interactions and identifying resistant and susceptible secondary metabolites or phytohormones are essential steps in maize crop improvement. These techniques offer valuable insights into the biochemical pathways involved in defense mechanisms, facilitating the development of resilient crop varieties and sustainable agricultural practices. Additionally, they provide detailed information on metabolites responsible for resistance and susceptibility to other major biotic stresses in maize (Table 2). Given the challenges posed by population growth, climate change, emerging pests and diseases, and the unsustainable use of agrochemicals, there is a growing urgency in adopting advanced genetic and metabolomics techniques to enhance crop resilience and reduce reliance on chemical inputs [14].

This review explores the potential of metabolomics to revolutionize the management of FAW infestations in maize. It delves into the current understanding of maize metabolism in response to pest attacks, the advancements in metabolomic technologies, and the practical applications of metabolomic data in enhancing pest management strategies. Through a comprehensive analysis of existing literature and recent research findings, this review aims to highlight the critical role of metabolomics in developing innovative, sustainable, and effective solutions to protect maize crops from the devastating impact of FAW.

## 2. Fall Armyworm—A Major Pest

Originally native to the tropical and subtropical regions of the Americas, FAW predominantly targets maize crops during autumn. It was first reported outside the Americas in West Africa in January 2016 and had spread over 40 African countries by 2018, showcasing its rapid dispersal and adaptability. The pest continued its invasion, reaching Asia, and was first reported in Karnataka, India in May 2018 [11,18]. FAW’s early emergence in the crop life cycle, as well as its voracious feeding habits, aggressive behavior, high reproductive rate, rapid migration and broad host range, and the severe and irreversible damage it inflicts on crops, establish it as a primary pest on maize, followed by sorghum and millets [1,5].

The impact of FAW infestation has been linked to estimated yield losses in maize ranging from 12 to 58% [19,20,21]. This review provides an overview of the utilization of metabolomics and metabolite responses to combat FAW and other major lepidopteron pests, aiming to minimize crop losses and enhance resistance through breeding strategies informed by metabolomics studies.

### Why Metabolomics?

Grain yield, the paramount agronomic characteristic, declines under FAW infestation, primarily due to significant foliar damage. This damage not only results in direct losses of photosynthesis, but also disrupts the normal functioning of the remaining leaf tissue. Breeders prioritize various strategies to address this challenge.

The following are the most common approaches to combat FAW:i. Biological control [22];ii. Conventional/native genetic resistance [11];iii. Transgenic resistance [23];iv. Cultural control [19];v. Habitat management [24].

Traditional management strategies for controlling *Spodoptera frugiperda* primarily rely on chemical insecticides, biological control agents, and cultural practices [11,25]. Chemical insecticides, including synthetic pyrethroids, organophosphates, and neonicotinoids, are widely used due to their immediate efficacy. However, their overuse has led to several challenges, such as the development of insecticide resistance in FAW populations, environmental contamination, and adverse effects on non-target organisms, including beneficial insects and soil health. Biological control involves the use of natural predators, parasitoids, and pathogens to suppress FAW populations. While environmentally friendly, these biological agents often require specific conditions to be effective and may not provide rapid control. Cultural practices, such as crop rotation, intercropping, and the use of resistant maize varieties, aim to disrupt the pest’s life cycle and reduce its population density. In a study conducted by Kasoma et al. [26], hybrids such as CML346/EBL16469, ZM4236/CML545-B, CML346/CZL1310c, CML334/EBL173782, and CML545-B/EBL169550 were identified for their favorable specific combining ability estimates concerning grain yield, days-to-50% anthesis, days-to-50% silking, and resistance to FAW leaf and cob damage. However, these methods can be labor-intensive, and may not be sufficient on their own to manage severe infestations.

In contrast, metabolomics offers a novel and holistic approach in managing FAW infestations by leveraging the plant’s inherent biochemical responses. Metabolomics involves the comprehensive profiling of metabolites within maize plants to understand their metabolic changes in response to pest attacks. This approach can identify specific metabolites associated with resistance or susceptibility to FAW. For instance, certain secondary metabolites, such as phenolic compounds, terpenoids, and alkaloids, may be upregulated in resistant maize varieties and play a role in deterring pests or reducing their survivability. By pinpointing these biochemical markers, researchers can develop maize varieties with enhanced natural defenses, reducing the reliance on chemical insecticides. Moreover, metabolomic insights can lead to the discovery of new bioactive compounds that could be formulated into biopesticides, offering a targeted and environmentally benign alternative to conventional chemicals.

## 3. Metabolomics Approaches to Unveil the Plant’s Chemical Orchestra

### 3.1. GC-MS and LC-MS

In the field of metabolomics, gas chromatography–mass spectrometry (GC-MS) stands out as a powerful analytical technique employed to separate and identify plant metabolites (Table 3)**.** In this method, molecules undergo separation based on their physical characteristics as they traverse through a heated column. Subsequently, these molecules enter the mass spectrometer where they are bombarded with electrons, resulting in the generation of unique fingerprints expressed as mass-to-charge ratios (*m*/*z*). By comparing these *m*/*z* values with existing databases, researchers can identify small metabolites such as amino acids, fatty acids, sugars, and organic acids, as well as unknown metabolites [27]. However, GC-MS is limited to detecting volatile compounds in plants.

In contrast, liquid chromatography–mass spectrometry (LC-MS) emerges as a valuable tool in plant metabolomics, complementing GC-MS by offering exceptional capabilities in interpreting complex metabolite profiles, e.g., as given in Table 4. LC-MS excels in profiling high-molecular-weight metabolites, heat-labile compounds, and unstable functional groups without the need for volatilization [16]. It is particularly adept at sensitively profiling lipids, sterols, and secondary metabolites, facilitating tandem mass spectrometry (MSn) for structural elucidation.

### 3.2. Other Techniques

The comprehensive characterization of communications is made possible by Fourier Transform Ion Cyclotron Resonance–Mass Spectrometry (FTICR-MS), which offers unmatched sensitivity and resolution. For the evaluation of ionic metabolites in plants, capillary electrophoresis–mass spectrometry (CE-MS) now provides an analytical avenue, enabling rapid separation without the necessity for derivatization. Despite its limitations in terms of sensitivity and reproducibility, CE-MS proves advantageous for use in focused analyses of ionic metabolites in plant metabolomics [37]. Matrix-Assisted Laser Desorption/Ionization Mass Spectrometry (MALDI-MSI) allows for both qualitative and quantitative analyses of metabolites in plant tissues. It provides spatial distribution information as well as compatibility with a variety of mass analyzers. The characterization of natural products in plants, metabolic fingerprinting, and biomarker identification are all made easier by MALDI-MS. To improve the capabilities of mass spectrometry (MS) in metabolite analysis, ion mobility spectroscopy (IMS) is being used frequently in plant metabolomics. Based on size and shape, IMS can differentiate between isomers. Apart from the potential improvements in data analysis and database development, IMS-MS offers advantages in metabolite identification, isomer separation, and peak capacity. Nuclear magnetic resonance (NMR) spectroscopy can be used to obtain quantitative and qualitative information on plant metabolites without sample derivatization or chromatographic separation. NMR-based metabolite profiling contributes to a thorough understanding of plant metabolism by providing benefits regarding sample preparation, repeatability, and non-destructiveness [37]. The types of metabolomics techniques are given in Figure 2.

Plants have an intricate defense system that combines molecular, enzymatic, and chemical processes to fend off herbivores. By identifying particular pathways that are activated in response to herbivore attacks, the researchers could gain insights into the roles of reactive oxygen species (ROS), secondary metabolites, antioxidant enzymes, phytohormones, biochemical and molecular aspects of how maize plants defend against infestation by the FAW [38]. This paper focuses on secondary metabolites and their role in resistance against FAW and other major lepidopteron pests.

### 3.3. Biochemical Reactions

The primary responses of plants to biotic stress relate to ROS generation and H_2_O_2_ concentrations. ROS are molecules produced by plants in response to herbivory. They can help plants communicate with each other, allowing them to coordinate their defense against herbivores. Overall, ROS plays an important role in the defense mechanisms of plants against herbivory. When the ROS concentration is higher, it also damages the cell membrane of the plants. Accordingly, in the study conducted by Yang et al. [38], herbivore attacks often showed increased ROS production and H_2_O_2_ concentration in plants. Maize cultivars ZD958 and JG218 recorded higher levels of hydrogen peroxide after FAW infestation, which could be associated with ROS production, while the cultivar JG218 was more susceptible than ZD958, as it showed greater malondialdehyde (MDA). MDA accumulation in infected leaves is indicative of increased ROS production, which leads to cell membrane damage due to increased oxidative stress [39,40,41]. The activation of antioxidant defense mechanisms in reaction to FAW infestation was also indicated by the upregulation of superoxide dismutase (*SOD*) (peroxidase (*POD*) and catalase (*CAT*) activities [38]. ZD958’s higher enzyme activity suggested greater resistance [42,43,44,45]. Plants’ rapid responses to insect damage are largely dependent on peroxidases [46,47]. Substantial progress has been made in crops such as grapes and pigeonpea, respectively; higher peroxidase expression has been reported in response to a variety of environmental stressors, such as cotton bollworm and non-adapted strains of two-spotted spider mites [48]. Variations in the expression of genes have been related to the metabolism of amino acids, including those for phenylalanine, glycine, serine, threonine, tyrosine, tryptophan, and cyano amino acid [38]. These amino acids are precursors or intermediates of a broad range of defensive secondary metabolites, such as auxin, JA, and indole, which can initiate antiherbivore defenses [49,50,51].

## 4. Primary and Secondary Metabolites

Primary metabolites are necessary chemicals that act as energy sources and cellular building blocks. These are essential for the survival, growth, and development of plants. Secondary metabolites play specific roles in defense against pathogens and herbivores, pollinator attraction, and UV protection, and are not required for basic biological processes [27]. Fraenkel. [52] mentioned that secondary metabolites play a complex and dual role in plant–insect interactions. They can act as attractants for specialized pest (especially monophagous insects) that have evolved to utilize specific plant species, while simultaneously serving as repellents for generalists or non-adapted insects (especially polyphagous insects). Secondary metabolites play versatile roles in ecological interactions by serving as both repellents to deter herbivores and pathogens and attractants to draw in beneficial organisms, such as pollinators, seed dispersers, and natural enemies of pests. Classifications of secondary metabolites are given in Figure 3. Protective substances such as plant secondary metabolites are crucial for interactions between plants and their surroundings [53]. Among these secondary metabolites, polyphenols stand out due to their extensive distribution and significant contribution to improving plant resistance to stress [43]. Due to their structural characteristics, polyphenols, which include phenolic acids, flavonoids, stilbenes, and lignans, actively shield plants from herbivores.

### 4.1. Phenolic Compounds

Phenolic compounds are a group of small molecules characterized by structures with at least one phenol unit. They are produced in plants via the shikimic acid and phenylpropanoid pathway as secondary metabolites, and are generally involved in plant adaptation to environmental stress conditions [54]. Based on their chemical structure, phenolic compounds can be divided into subgroups such as phenolic acids, flavonoids, tannins, coumarins, lignin, quinones, stilbenes and curcuminoids [55]. Types of secondary metabolites are given in Figure 4. Most phenolic compounds are believed to be by-products of the metabolism of the amino acid phenylalanine.

Abundant studies have also emphasized the role of polyphenols in providing resistance to both biotic and abiotic stresses [56]. Phenolic acids are a diverse class of plant polyphenols and are the most widely studied too, and they are produced through shikimic acid by the phenylpropanoid pathway. Phenylpropanoids contribute to all aspects of plant responses towards biotic and abiotic stimuli. They are not only indicators of plant stress responses upon variations in light or mineral treatment, but are also key mediators of the plant’s resistance towards pests [57]. Many phenylpropanoid pathway-derived molecules act as physical and chemical barriers to pests and pathogens. Cinnamate 4-hydroxylase (C4H) is a cytochrome P450-dependent monooxygenase that catalyzes the second step of the general phenylpropanoid pathway. Desmedt et al. [58] identified piperonylic acid (PA) as an effective, broad-spectrum, and non-phytotoxic novel resistance inducer by exploring the phenylpropanoid pathway through the application of the cinnamic acid-4-hydroxylase (C4H) inhibitor. PA, a natural molecule bearing a methylenedioxy function, closely mimics the structure of trans-cinnamic acid. Using bioassays involving diverse pests and pathogens, the researchers demonstrated that transient C4H inhibition triggered systemic, broad-spectrum resistance in higher plants without affecting growth. Their study, conducted on tomato plants, confirmed that PA treatment enhanced resistance in both field and laboratory conditions, thereby illustrating the potential of phenylpropanoid pathway perturbation in crop protection and suggested that PA can be effectively used to control pest incidence in crops.

Quercetin, gallic acid, caffeic acid, syringic acid, p-coumaric acid, ferulic acid, and cinnamic acid were among the phenolics that have been found to exhibit high herbicidal and insecticidal properties against *Oxycarenus hyalinipennis* (Costa), a dusky cotton bug (*Oxycarenus laetus* [Kirby]), and duckweed, *Lemna minor* L. [59]. Likewise, chewing insects like *Helicoverpa armigera* (Hubner) and *Spodoptera litura* (Fabricius), when fed on cotton plants, metabolized compounds such as gallic acid, 4-cinnamic acid, p-coumaric acid, and salicylic acid, resulting in significant weight loss and mortality among the target insects [60]. In the same way, the study undertaken by Dowd and Sattler [61] revealed that phenolic acids *viz*., ferulic, vanillic, sinapic, and syringic acids in artificial diets inhibited the growth of FAW on sorghum. Researchers reported that phenolic compounds like p-coumaric acid and resveratrol lowered the larval weights of *Amsacta albistriga* (Walker) and *Spodoptera litura* [62]. The total phenolic and flavonoid contents in infested tissues were significantly higher than in healthy tissues, according to studies on cucurbits infested with melon flies and white cabbage infested with flea beetles and cabbage butterflies [44,63]. With this, maize plants infested with FAW exhibited a notable increase in total phenolic contents following the infestation, indicating that the plants had initiated a defense mechanism [38]. Together, the importance of polyphenols and associated enzymatic activities in plant defense against herbivores highlights the complexity of plant–herbivore interactions and the potential application of these mechanisms in agricultural practices to control pests, as well as the effects of phenolic compounds against other lepidopteran pests, all of which is tabulated in Table 5.

#### 4.1.1. Flavonoids

Flavonoids are small molecular secondary metabolites synthesized by plants with various biological activities. Among several secondary metabolites, flavanols, a subgroup of flavonoids, are one of the most widely distributed in the plant kingdom. The flavonoid (anthocyanin) biosynthesis pathway is illustrated in Figure 5. These molecules work as antioxidants, reduce reactive oxygen species (ROS) in plants, and cause detrimental effects on insect growth via alterations in feeding. Sorghum lines with lower flavonoid levels outperformed those reported in the study conducted with controls in terms of FAW growth, adding to the evidence supporting sorghum flavonoids’ protective function against FAW. Experts reported an increased flavonoid content in SC1345 plants in response to FAW attack, whereas phenolic accumulation was inhibited [76]. Sorghum flavonoids extracted from leaves added to an artificial diet and fed to insects have addressed FAW mortality [77]. Additionally, by virtue of making it harder for the FAW larvae to survive, sorghum 3-deoxy anthocyanidins have been sprayed on leaves of a susceptible maize line and have been observed to discourage herbivory growth [77]. Certain genes found to be involved in the flavonoid biosynthesis pathway, including two shikimate O-hydroxycinnamoyl transferase genes and one flavonoid 3′-monooxygenase gene, were induced to express themselves following herbivory [78]. According to research on flavonoid contents in FAW-affected maize plants, the plants may have launched a defense mechanism in response to infestation [38]. Flavonoids’ roles in other major lepidopteron pests are tabulated in Table 4.

#### 4.1.2. Tannins

The adaptability of herbivores to different host species also depends on plant morphologies, palatability, nutritional contents, and secondary metabolites. In this case, phenolic compounds are considered major secondary metabolites against herbivores. In various plant species, tannins serve as defensive compounds against insect herbivory, which has been seen to be active against several phytophagous pests, such as gypsy moth (*Lymantria dispar* L.), brown-tail moth (*Euproctis chrysorrhoea* L.), Winter moth (*Operophtera brumata* L.), and cowpea aphids (*Aphis craccivora* K.) [79]. Liu et al. [80] observed no noticeable differences in terms of the total phenol content between wheat and faba bean leaves, but the tannin content was significantly higher (6.19 times) in faba bean leaves compared with wheat leaves, and they also reported that combined analyses of the biological parameters of *Spodoptera frugiperda* on faba bean leaves suggested that lower nutritional quality and higher toxic tannin contents resulted in the low adaptability of *Spodoptera frugiperda* larvae. They also suggested that cultivars with high tannin contents showed resistance against polyphagous pests.

### 4.2. Terpenoids

Terpenoids, also known as isoprenoids, are naturally occurring organic compounds that are derived from the five-carbon compound isoprene and its derivatives, which are known by various names, such as diterpenes, terpenes, etc. While “terpenoids” and “terpenes” are sometimes used synonymously, terpenoids are distinguished by the presence of additional functional groups, the majority of which are oxygen-containing. The hydrocarbon terpenes are mixed with the approximately 80,000 compounds that comprise terpenoids. They are the largest class of secondary metabolites found in plants, accounting for about 60% of all known natural products. Herbivores produce plant volatiles called terpenoids, which can either attract or repel herbivore predators. Terpenoids are aerial messengers that can incite defense responses in nearby plants as well as in systemic, unharmed areas of the plant. Terpenoids are emitted either constitutively or induced in response to biotic [81,82,83,84] and abiotic [85,86] stresses. They are important members of the class of herbivore-induced plant volatiles (HIPVs) [87,88]. In maize the volatiles produced include a range of terpenes, which are likely produced by several of the terpene synthases (TPS) present. Terpenoid’s role against FAW was formulated in Table 3.

### 4.3. DIMBOA (2,4-dihydroxy-7-methoxy-1,4-benzoxazin-3-one)

DIMBOA is a naturally occurring hydroxamic acid, a benzoxazinoid and a powerful antibiotic present in maize, white, rye and related grasses. The benzoxazinoid biosynthetic pathway is illustrated in Figure 6. As a powerful biochemical defense mechanism, maize seedlings constitutively produce a wide variety of benzoxazinoid compounds in high abundance [34]. It was thought that the aforementioned compound was the source of the inhibitory effect seen against herbivorous insects. Flypaper wasp larvae are adapted to defend themselves against maize, and several benzoxazinoid compounds have been observed to differ in degrees of success [89]. HDMBOA-Glc, which has been shown to repel and suppress the growth of FAW larvae, was not the only benzoxazinoid compound found in high concentrations in maize cultivar Xi502 leaf tissues [89], and two extremely common breakdown products, M2BOA and 6-MBOA, have been found to be accumulated at higher levels in Xi502 than in B73 [90]. This is the pattern of benzoxazinoid glucosides. Benzoxazinoids have suppressed herbivore growth under iron-deficient conditions, and in the presence of chelated iron, enhance herbivore growth in the presence of free iron in the growth medium [91]. In response to infestation, DIMBOA levels were found to be sharply increased, suggesting a possible role for DIMBOA in plant defense. Compared to maize cultivar JG218, the resistant genotype, ZD958, was found to exhibit higher DIMBOA levels [34,92,93,94].

In maize, benzoxazinoid biosynthesis begins in the chloroplast resulting from the conversion of indole-3-glycerol phosphate (an intermediate of tryptophan biosynthesis) into indole, catalyzed by the indole-3-glycerol phosphate lyase benzoxazinoneless 1 (BX1). A subsequent stepwise introduction of four oxygen atoms by the P450 monooxygenases BX2, BX3, BX4, and BX5 leads to the formation of 2,4-dihydroxy-1,4-benzoxazin-3-one (DIBOA). DIBOA is a substrate for the UDP-glucosyltransferases BX8 and BX9, which convert the toxic compound DIBOA into the stable glucoside (Glc) form DIBOA-Glc. The 2-oxoglutaratedependent dioxygenase (2ODD) BX6 catalyzes the hydroxylation of DIBOA-Glc at C-7, followed by a methylation of the introduced hydroxyl group catalyzed by the O-methyltransferase BX7, yielding DIMBOA-Glc in the cytosol. An O-methylation reaction catalyzed by a group of three Omethyltransferases (BX10, BX11, and BX12) converts DIMBOA-Glc to 2-hydroxy-4,7-dimethoxy-1,4-benzoxazin-3-one glucoside (HDMBOA-Glc). BX13, a BX6-like 2-ODD, catalyzes the conversion of DIMBOA-Glc to 2,4,7-trihydroxy-8-methoxy-1,4-benzoxazin-3-one glucoside (TRIMBOA-Glc). TRIMBOAGlc can be O-methylated by BX7 to form 2,4-dihydroxy-7,8-dimethoxy-1,4-benzoxazin-3-one glucoside (DIM 2 BOA-Glc), which can be further methylated by the O-methyltransferase BX14 to generate 2-hydroxy-4,7,8-trimethoxy-1,4-benzoxazin-3-one glucoside (HDM 2 BOA-Glc). BX14 can also produce HDMBOA-Glc from DIMBOA-Glc. The benzoxazinoid glucosides (Bx-Glc) are stored in the vacuole, where they are protected from b-glucosidases located in the chloroplast and cell wall. Upon cell disruption (e.g., by herbivory), the Bx-Glc are exposed to the b-glucosidases, which cleave the glucosyl group, generating bioactive aglucons. Abbreviations: ER, endoplasmic reticulum; TRIBOA-Glc, 2,4,7-trihydroxy-1,4-benzoxazin-3-one glucoside. (Source: KEGG).

### 4.4. Glucosinolates 

Benzoxazinoids (BXs) are widely distributed phytoanticipins among the Poaceae family. It is commonly assumed that BX-glucosides are hydrolyzed by plastid-targeted β-glucosidases upon tissue disruption, resulting in the release of biocidal aglycone BXs. Glucosinolates, on the other hand, are found mainly in the Brassicaceae family (mustards and cabbages). They also act as phytoanticipins, stored in an inactive form and hydrolyzed by the enzyme myrosinase upon tissue damage. Similarly, glucosinolates (GSs), a class of thioglucosides containing nitrogen, are natural products found in plants, particularly within the order Capparales and primarily represented by the family Brassicaceae [95]. Intact glucosinolates have been found to be linked to very little biological activity, but their breakdown products play a crucial role in plant defense. The levels of glucosinolates vary among plant organs, developmental stages, and individual species, and are influenced by both biotic and abiotic factors. The susceptibility of plants to insect pests is determined by the type and quantity of glucosinolates present. When plant tissue is damaged, various biotic or abiotic factors cause the hydrolysis of glucosinolates, resulting in the production of isothiocyanates, thiocyanates, and nitriles. These breakdown products have differing effects on pests; isothiocyanates and nitriles stimulate specialist pests but are generally repellent to generalist pests [96]. Lv et al. [97] reported on the influence of individual glucosinolates, which were found to significantly affect the feeding behaviors of pests such as the bertha armyworm (*Mamestra configurata* [Walker]), and they mentioned that glucosinolates can be synthesized de novo in response to microbe or insect attack, as seen with phytoalexins, or produced constitutively and stored in an inactive form in the plant cell. Well-characterized examples of phytoanticipins, such as glucosinolates, can be hydrolyzed by endogenous β-thioglucoside glucohydrolases known as myrosinases. This hydrolysis is part of the plant’s strategy of defending itself from harmful organisms, utilizing glucosinolates and their breakdown products to protect Brassicaceae plants grown worldwide [98].

## 5. Feeding Impacts of Metabolites for FAW

Larval feeding affects maize plants’ capacity to produce defensive compounds. Targeted metabolites such as linoleic acid have been increased significantly as a result of larval feeding, which has negatively impacted FAW larvae [37]. When compared to undamaged control plants, analyses of volatile organic compounds have reported significantly higher concentrations of several compounds in maize plants induced by larval feeding [37]. In a variety of plant–insect systems, insect cues can modify particular defense responses, and their impacts differ. When chewing on plants, insects release signals such as saliva and regurgitant. Depending on the intricate relationships between plants and insects, these cues may have advantageous or disadvantageous consequences. FAW saliva and regurgitation have been identified to trigger defense responses in maize [99]. According to Jones et al. [100], FAW has a heat-stable effector in its caterpillar regurgitant that can reduce the emission of volatile compounds in maize plants. The application of FAW regurgitant has increased the extent of the use of flavonoids as potent defensive components and activated several genes encoding enzymes required for flavonoid biosynthesis [99].

## 6. Metabolomic Response for Different Strains of FAW

The sympatric distribution of FAW in North and South America is characterized by two genetically and behaviorally distinct strains [101]. It has been reported that the “rice” strain (R strain) primarily infests rice, alfalfa, pasture, and forage grasses, while the “corn” strain (C strain) preferentially damages maize, sorghum, and cotton [6]. Although the morphological characteristics of these FAW “host strains” are identical, polymorphisms in the mitochondrial cytochrome oxidase subunit I (*COI*) gene are able to reliably differentiate and identify the C and R strains according to their haplotypes [102]. Maize was found to exhibit a stronger defense response than the rice strain in response to the herbivory of the corn strain because of the higher accumulation of Jasmonic acid (JA) and Jasmonoyl Isoleucine (JA-Ile) Jasmonates have been well documented in relation to maize’s defense against insect attack, particularly from other *Spodoptera* species [103]. An upregulation of ZmAOS2 transcripts and an increased accumulation of jasmonates have been identified as indicators of elevated defense response. In the case of the rice strain’s herbivory, maize plants produce more defense hormones and are found to accumulate more defensive benzoxazinoids. Additional investigations using untargeted metabolite analysis have stated that, in the first hour of herbivory feeding, the herbivorous rice strain showed a greater accumulation of various fatty acid derivatives, such as 18:3 -2OH and 18:3 -3OH, than the herbivorous maize strain [104]. According to the experts, plants store fatty acids and their derivatives in response to stress, especially following herbivory and wounding [105,106,107]. By making the intestinal lumen more acidic, shikimic acid lowers intestinal proteolytic activity in insects. However, the bacteria in these insects’ stomachs enable them to metabolize the chemical and lessen its effects [108]. It appears that the C strain and R strain behave quite similarly when feeding on this plant because only two substances, glycerol monostearate and 2-isopropylaminoethanol, were differentially abundant between the two strains, and could be observed in the midgut of C strain larvae. The corn strain larvae’s midguts contain more shikimic acid than the rice strain larvae do, suggesting that the corn strain is less capable of processing this metabolite. Dietary effects indicate that different FAW strains have distinct digestive metabolisms. Furthermore, it has been discovered that marker metabolites could provide insight into the mechanisms behind host adaptation [104].

## 7. Future Prospects

Recent advances in transcriptomic and metabolomic technologies facilitate a novel trend of integrated “omics”. Transcriptome analysis is an efficient method for the large-scale screening of genes associated with specific traits in an organism. Metabolites, on the other hand, are direct agents of plant defense against insect feeding and respond immediately to pest attacks. Additionally, metabolites are either direct or indirect products of gene expression [78,109]. Integrating these technologies would be useful for studying the metabolic changes in plants in response to herbivore infestation. The pathway linked to amino acid metabolism, which includes glutathione, tryptophan, tyrosine, cysteine, methionine, phenylalanine, alpha-linolenic acid, diterpenoid, and linoleic acid metabolism, has been observed to contain 18 Differentially expressed genes (DEGs) and 12 Differently expressed metabolites (DEMs). According to Li et al. [78], there were three DMs and four DEGs linked to the biosynthesis of phenylpropanoid, suggesting that increasing those DEMs and DEG could be sufficient to elevate insect resistance. In the case of tomatoes, trichomes are multicellular structures, with types I, IV, and VI subjected to the most research. Types I and IV are the main sources of acyl sugars. Type VI glandular trichomes, which have high concentrations of volatile organic compounds like flavonoids, phenylpropanoids, sesquiterpenes, and monoterpenes, offer indirect defense against insects. These substances attract parasitic wasps and other carnivorous plant predators [110]. Combined transcriptome and metabolome data have revealed that DEGs and DEMs were significantly enriched in the flavone biosynthesis (ko00941) and phenylpropanoid biosynthesis (ko00940) pathways. More specifically, ρ-coumaric acid (ρ-CA), found to be catalyzed by *PAL*, *C4H*, *4CL*, and Cinnamyl Alcohol Dehydrogenase (*CAD*), which generates ρ-coumarin alcohol (C02646), has been identified as the precursor to multiple downstream metabolic pathway branches. The co-expression of Cinnamate-4-Hydroxylase (*C4H*) and 4-CoumarateLigase (*4CL*) with Phenylalanine Ammonia-Lyase (*PAL*) satisfied the requirements of the second and third stages of the phenylpropanoid metabolic pathway. Two *C4H*s and four *4CL*s have been identified in SH tomatoes, and all of them were upregulated, indicating that *C4H* and *4CL* are likely important regulatory genes for SH in fending off insect invasion. Among them, *4CL* is a crucial branch-point enzyme linked the phenylpropanoid metabolism pathway and downstream metabolic pathways [109]. It has been stated that FAW herbivory enhances the biosynthetic pathways in Kanlow grass, resulting in the synthesis of multiple secondary metabolites. The buildup of terpenoids and other secondary defense metabolites on Kanlow plants has a detrimental effect on FAW development and growth in comparison to summer plants. However, it has been discovered that increased terpenoid compound production has insecticidal effects on a variety of crops [111,112]. Li et al. [78] focused on the genes involved in the defense against insect herbivory in sugarcane through transcriptome analysis combined with metabolome analysis. The results reveal that the defense response of plants to pests is a complex process, involving changes in the expression of a large number of genes and metabolites associated with hormone biosynthesis and defense mechanisms. This includes genes related to secondary metabolism, peroxidases, GSTs, and heat shock proteins. The defense mechanism involves various transcription factors (TFs) such as Myeloblastosis, WRKY, Ethylene Response Factor, and signal transduction through different phytohormones such as salicylic acid and jasmonic acid. According to this, it is possible to identify the genes in charge of producing defense metabolites by combining transcriptomics and metabolomics data, which makes metabolomics-based breeding successful. 

## 8. Conclusions

The review underscores the significance of metabolites in cereals facing various biotic stresses, notably FAW. It examines crucial metabolites and emerging metabolic pathways essential for plant defense. Consequently, in programs aiming to enhance transgenic crops, altering metabolite biosynthetic pathways emerge as a promising strategy. Metabolomics contributes to sustainable pest management by pinpointing potential targets for breeding FAW-resistant maize varieties by identifying specific metabolic signatures linked to FAW resistance. Through the integration of metabolomics with other omics techniques and advanced computational methods, researchers can delve deeper into the molecular mechanisms governing FAW–plant interactions. This integration promises to pave the way for more innovative and targeted FAW control strategies.

## Figures and Tables

**Figure 1 plants-13-02451-f001:**
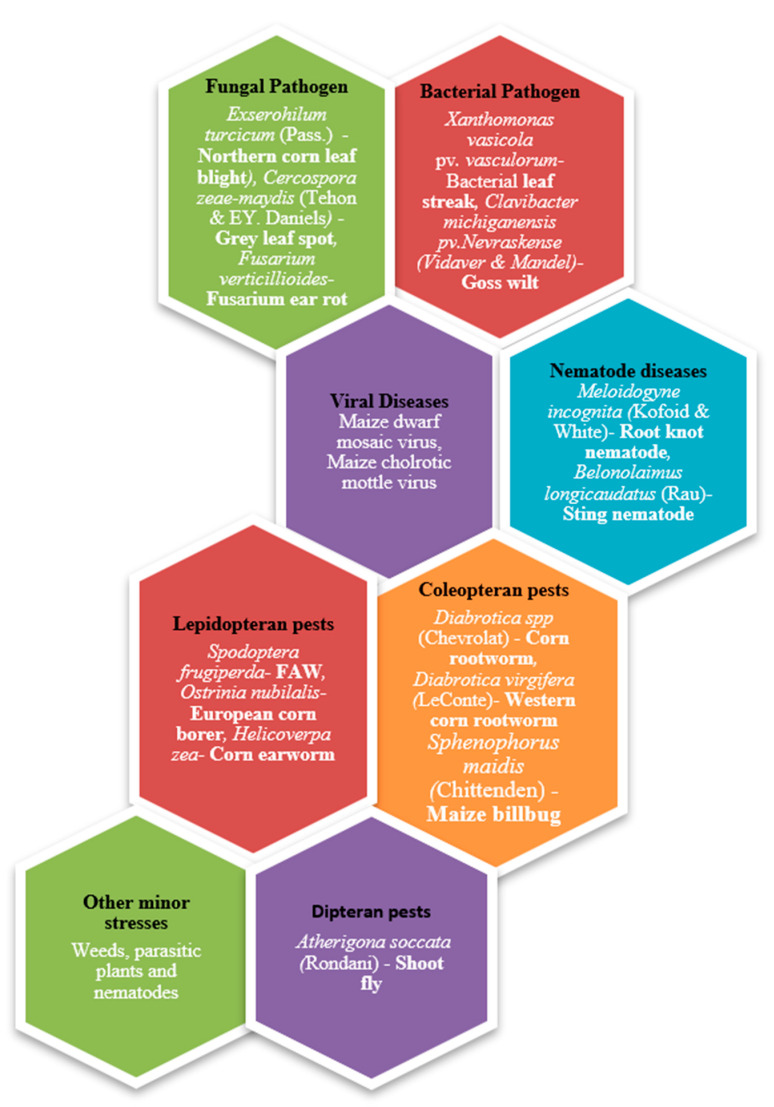
Types of biotic stress that maize plants encounter (categorized based on the type of organism causing the stress).

**Figure 2 plants-13-02451-f002:**
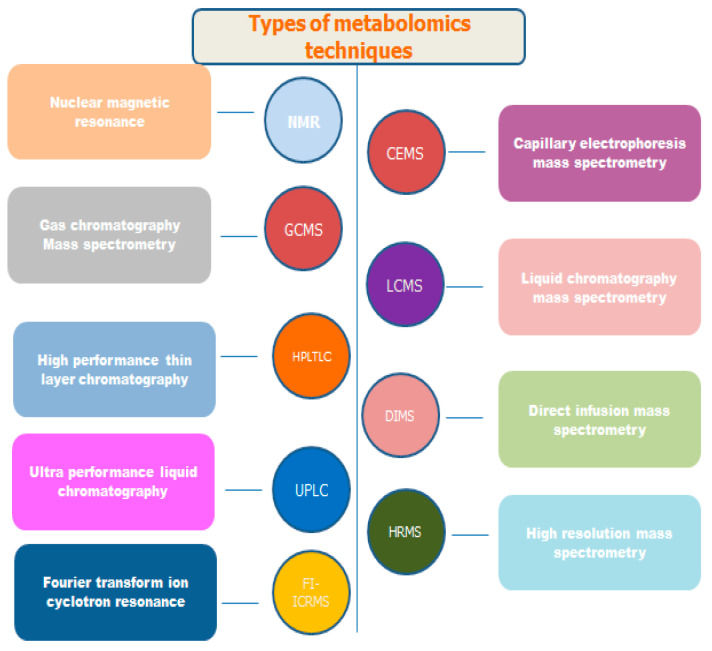
The major techniques used in metabolomics.

**Figure 3 plants-13-02451-f003:**
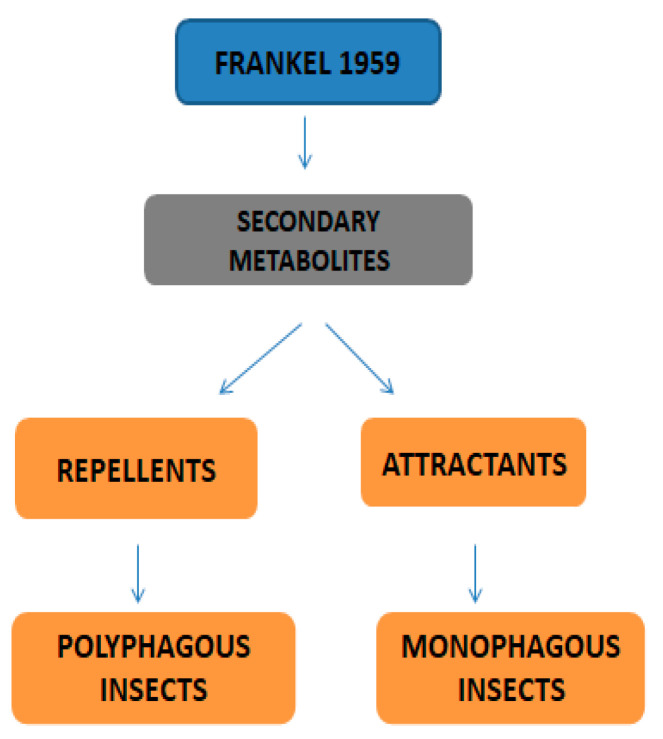
Classifications of secondary metabolites.

**Figure 4 plants-13-02451-f004:**
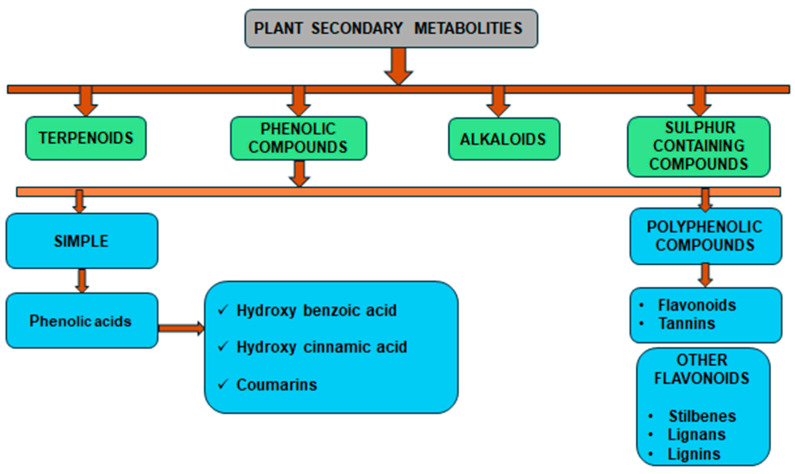
The diverse range of secondary metabolites employed by plants for defense against herbivores, pathogens, and other environmental threats.

**Figure 5 plants-13-02451-f005:**
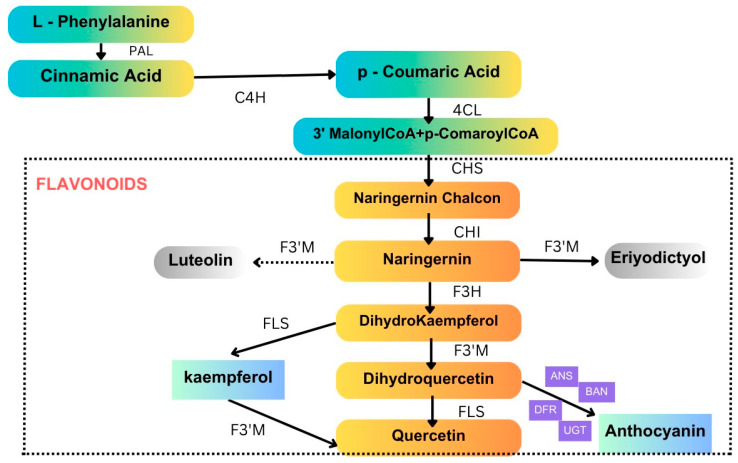
Flavonoid (anthocyanin) biosynthesis pathway. Anthocyanins are a type of flavonoid with specific properties that contribute to plant defense. PAL, phenylalanine ammonia lyase; C4H, cinnamate 4-hydroxylase; 4CL, 4-coumarate: CoA ligase; CHS, chalcone synthase; CHI, chalcone isomerase; F3H, flavanone 3-hydroxylase; F3′M, flavonoid 3′-monooxygenase; FLS, flavonol synthase; DFR, dihydroflavonol 4-reductase; ANS, anthocyanidin synthase; BAN, banyuls; UGT, UDP-glucosyl transferase 78D3. (Source: KEGG).

**Figure 6 plants-13-02451-f006:**
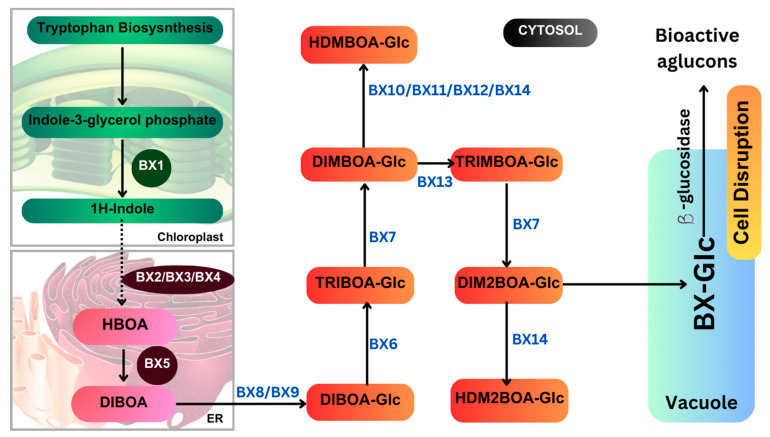
Benzoxazinoid biosynthetic pathway.

**Table 1 plants-13-02451-t001:** Difference between targeted and Non-targeted metabolomics.

Feature	Targeted	Non-Targeted
Focus	Particular, a predetermined group of metabolites	The broad spectrum of unknown and known metabolites
Analytical technique	Usually LC-MS/MS, GC-MS	Various techniques like LC-MS, GC-MS, NMR
Data complexity	Lower	Higher, requires advanced data processing and analysis tools
Quantification	Absolute quantification is possible for known metabolites	Relative quantification, identification of unknown metabolites
Applications	Biomarker discovery, metabolic pathway analysis, targeted gene expression studies	Metabolite discovery, phenotypic characterization, plant stress response analysis
Cost	Less expensive due to focused analysis	More expensive due to broader analysis and complex data processing

**Table 2 plants-13-02451-t002:** Metabolites for other biotic stress.

Other Major Biotic Stress	Metabolites/Compounds	Role/Function	Role in Resistance/Susceptibility	Reference
*Helicoverpa zea*	Homogalacturonan breakdown	Cell wall modification	Weakening the cell wall and hindering insect feeding	[15]
	Epicuticular wax formation	Cuticle reinforcement	Providing a physical barrier against insect penetration	
	Gibberellic acid synthesis 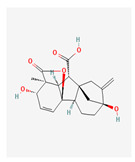	Plant growth regulation	Modulating plant growth to deter insect infestation	
	Fatty acid production	Lipid metabolism	Contributing to the formation of a physical barrier and defense signaling	
	Cellulose biosynthesis	Cell wall synthesis	Strengthening cell walls and hindering insect feeding	
	Phospholipase activity	Lipid metabolism	Contributing to the formation of a physical barrier and defense signaling	
	DIMBOA glucoside biosynthesis 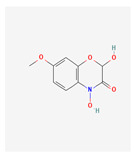	Benzoxazinoid synthesis	Antibiosis, producing toxic substances for insect deterrence	
	Coumarin biosynthesis 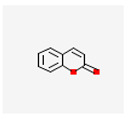	Secondary metabolite synthesis	Antibiosis, producing toxic substances for insect deterrence	
	Anthocyanin biosynthesis 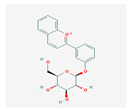	Secondary metabolite synthesis	Antibiosis, producing toxic substances for insect deterrence	
*Fusarium verticillioides*	Aminoacyl-tRNA biosynthesis	Protein synthesis	Enriched in both resistant and susceptible RILs	[16]
	Cysteine metabolism 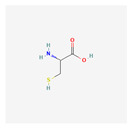	Sulfur-containing amino acid metabolism	Enriched in both resistant and susceptible RILs	
	Methionine metabolism 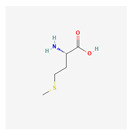	Sulfur-containing amino acid metabolism; detoxification	Enriched in resistant RILs; potential accumulation of detoxification metabolites	
	Arginine metabolism 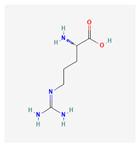	Nitrogen metabolism	Enriched in both resistant and susceptible RILs	
	Proline metabolism 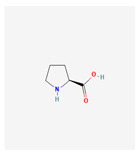	Osmoprotectant; stress response	Enriched in both resistant and susceptible RILs	
	Glutathione metabolism 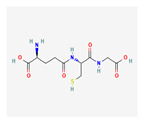	Antioxidant; detoxification	Enriched in both resistant and susceptible RILs	
	Lipid metabolism (including phosphatidylcholines)	Membrane integrity; reactive oxygen species (ROS) scavenging	Changes more significant in resistant RILs at 10 days after infection (dat)	
	Auxin homeostasis	Plant hormone regulation	Higher accumulation in resistant RILs	
	Phenylpropanoid pathway	Secondary metabolite synthesis	Upregulated in resistant RILs	
	Isoquinoline metabolism 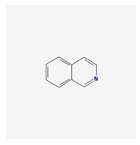	Secondary metabolite synthesis	Differential accumulation at 10 dat, potential involvement in resistance	
	Octadecadienoic acid derivative 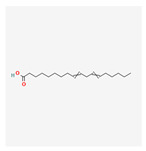	Lipid metabolism; signaling molecule	Differential accumulation at 10 dat, potential involvement in resistance	
	Sinapic acid 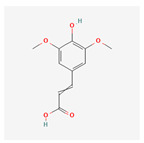	Phenylpropanoid compound	Differential accumulation at 10 dat, potential involvement in resistance	
	Ferulic acid 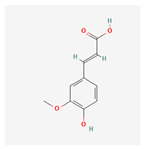	Phenolic compound; antioxidant	Discriminant at 3 dat, potential involvement in early cell damage response	
	Benzoxazinoid metabolism 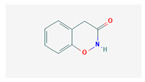	Secondary metabolite synthesis; insecticidal properties	Discriminant at 3 dat, potential involvement in early cell damage response	
*Puccinia sorghi*	Phytohormones (e.g., ethylene, abscisic acid, jasmonic acid)	Regulation of plant defense responses	Fine-tuning defenses mediated by JA against herbivores	[17]
	Alkaloid compounds 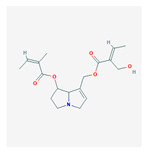	Likely involved in defense mechanisms	Accumulation observed after prolonged feeding	
	Benzoxazinoids and kauralexins 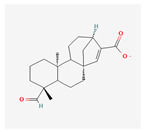	Antibiotic-acting compounds	Not increased after prolonged feeding	
	Amino acids	Substrate for the biosynthesis of defense compounds	Control accumulation of likely alkaloid compounds	
	Glutathione-related compounds (e.g., L-Cys-Gly, reduced glutathione)	Antioxidant and detoxification functions	Lower levels observed due to the higher expression of detoxification-related enzymes	
	Dehydroascorbic acid (DHA) 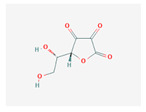	Oxidized form of ascorbic acid	Accumulates due to lower levels of reduced glutathione	
	Monodehydroascorbate reductase	Enzyme involved in maintaining ascorbic acid levels	Overexpressed to control ascorbic acid levels	

**Table 3 plants-13-02451-t003:** Terpenoids’ role against FAW.

Crop	Pest	Instrument	Compounds Induced by Herbivory	References
*Zea mays* L.	*Spodoptera frugiperda*	-	Monoterpene, Monoterpene alcohols, Homoterpenes, Sesquiterpenes (E)-β-Caryophyllene	[28]
*Tagetes erecta* L.	*Spodoptera frugiperda*	FITR	Terpenoids, tannins, Phenols, alkaloid, flavanol	[29]
*Hyptis marrubioides* & *Ocimum basilicum* L.	*Spodoptera frugiperda*	GC-MS	Linalool, α-thujone, 1,8-cineole	[30]
*Zea mays* L.	*Spodoptera frugiperda*	-	Monoterpene volatiles β-myrone, linalool	[31]
*Panicum virgatum* L.	*Spodoptera frugiperda*	GC-MS	Monoterpenes, sesquiterspens	[32]

**Table 4 plants-13-02451-t004:** Flavonoids’ role in other major lepidopteran pests.

Crop	Insect	Instrument	Metabolties Studied/Identified	Resistance Compounds	Reference
*Glycine max* L.	*Spodoptera litura*	LRLC-MS + HPLC	Diadzein, 4,7, dihydroxy flavone, genistein, kaemferol, apigenin, forrononetin, soyabean flavonoid aglycones	Isoflavones	[33]
*Glycine max* L.	*Spodoptera litura*	HPLC	Seven isoflavonoods, cyclitol, two sterol derivatives, three triterpenoids	Isoflavonoid, Diadzein	[34]
*Cajanus cajan* L.	*Helicoverpa armigera*	LC-MS	Total protein content	Flavonoid Isoorientin	[35]
*Amaranthus cruentus* L.	*Spodoptera litura*	-	-	Flavonoid glycosides, vitexin, vitexin-2	[36]

**Table 5 plants-13-02451-t005:** Phenolic compounds responsible for resistance to other lepidopteran pests.

Crop	Insect	Instrument	Metabolites Studied/Identified	Resistance Molecules	Reference
*Solanum lycopersicum* L.	*Spodoptera litura*	TLC, HPLC, FTIR	P-Kaempferol, rutin, caffeic acid, p-courmaric acid, Flavonoid Glycoside	Kaempferol, coumaric acid	[64]
*Acacia nilotica* L.	*Spodoptera litura*	HPLC, NMS-MS	-	Catechin. Chlorogenic acid, umbelliferone	[65]
*Acacia nilotica* L.	*Spodoptera litura*	UHPLC	-	11 phenolic compounds	[66]
*Acacia nilotica* L.	*Spodoptera litura*	UHPLC	-	Ferulic acid	[67]
*Acacia nilotica* L.	*Spodoptera litura*	UHPLC	-	Pyrogallol	[68]
*Acacia nilotica* L.	*Spodoptera litura*	UHPLC	-	Gallic acid	[67]
*Acacia nilotica* L.	*Spodoptera litura*	UHPLC	-	Ellagic acid	[69]
*Capsicum annum* L.	*Spodoptera litura*	HPLC	-	Protein carboxyl content and acetyl cholinesterase activity	[70]
*Acorus calamus* L.	*Spodoptera litura*	HPLC	-	Caffeic acid	[71]
*Arachis hypogaea* L.	*Spodoptera litura*	HPLC	Phenols	Cholrogenic, syringic, quercitin, ferrulic acid	[72]
*Zea mays* L.	*Spodoptera litura*	-	-	Alpha amylase and higher content of phenolic compounds	[73]
*Zea mays* L.	*Spodoptera litura*	-	-	Total phenols and tannins	[74]
*Zea mays* L.	*Spodoptera litura*	UFLC	-	P-comaric acidFerulic acid	[75]

## Data Availability

The original contributions presented in the study are included in the article, further inquiries can be directed to the corresponding author.

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
