# Peer review of "Exploring Metabolomics to Innovate Management Approaches for Fall Armyworm (*Spodoptera frugiperda* [J.E. Smith]) Infestation in Maize (*Zea mays* L.)"

_plants, 2024, doi:10.3390/plants13172451_

Round 1

Reviewer 1 Report

Comments and Suggestions for Authors

A brief summary

Fall armyworm (FAW) Spodoptera frugiperda (J.E. Smith) is a maize insect pest that has become a major global problem and causes extensive yield losses. The authors of this manuscript put together a nice review exploring potential of metabolomics approaches in addition to conventional methods to manage FAW. Besides abstract, the manuscript started with introduction followed with FAW – a major pest, metabolomics approaches to unveil the plant’s chemical orchestra (the author presented this part with possible techniques), primary and secondary metabolites, feeding impacts of metabolites for FAW, metabolomic response for different strains of FAW, and ended with future prospects and conclusion. Tables and figures enhanced the contents of the manuscript and provided great examples for the subject the authors attempted to convey. Overall, the manuscript provided an excellent review and summary of various metabolomics potential in combatting FAW. Having broken down the body of manuscripts into subtopics helped readers understand the points that authors attempted to make.

Specific comments:

There are many inconsistencies in the document, for example, spelling error of common name of pest. Also, I found major errors on technical terms in one of the figures. I may have missed some errors but I tried to point out below. Authors need to go through thorough review in correcting them.

Title (page 1)

Line 2

Incorrect spelling for FAW scientific name; it should be Spodoptera frugiperda, NOT Spodopthera …

Introduction (page 2)

Line 62

Fall army worm à it should be fall armyworm with lower case for “f” since it is in the middle of the sentence. Also, armyworm is ONE word, not two words.

You need to check the entire document for this.

Fall Armyworm – a Major pest (page 3)

Line 103

Check the publisher guidelines: Is the title needs to be sentence case or lower case?

Line 123

Should add “the most common approaches” to combat FAW. They are not the only approaches.

Metabolomics approaches to unveil … (page 4-5)

Lines 158-205

Could authors give one example of each technique by referencing each technique work? If this already done in one of the tables, refer the table.

Lines 175-179

Your subtitle only said “GC-MS and LC-MS”. Is CE-MS a different kind of LC-MS? If so, please state. If not, the subtitle should be adjusted to reflect of such technique.

Line 188

MALDI-MSI à should be MALDI-MS àdelete “I”

Line 206 (page 5)

“:” should be deleted

Line 279 (page 6)

One example of inconsistency.

The authors spelled out Helicoverpa armigera yet abbreviated S. litura. If other species was spelled out, then “S. litura" needs to be spelled out as well.

Lines 357-359 (page 7)

Is Xi502 a maize cultivar? If so, please indicate as such to clarify since it was not clear.

Line 407 (page 8)

What are CS and RS? Although both are listed in the abbreviation list, it should be spell out for the first time then abbreviate throughout the document.

Table 2 (page 10)

Line 476

Another inconsistency example here.

Check with other tables and decide whether you would like to use common name or scientific name of the insect pest. If use common name, correct spelling is corn earworm (ONE word for earworm, not two words). If use scientific name, it is Helicoverpa zea.

Table 3 (page 13)

Line 477

Under “insect” column: incorrect spelling for Spodopthera, it should be Spodoptera, i.e. without “h”.

Table 4 (page 14)

Line 480

Same issue as Table 3.

Under “crop” column: correct spelling is “soybean” (without “a” between y and b, not soyabean). Also, either use common name or scientific name à be consistent with other tables.

Table 5 (page 14)

Line 481

Under “pest” column: use common name or scientific name à be consistent with other tables.

Figure 1 (page 15)

Lines 484-486

Correct the following hexagons:

-      1.  Corn rootworm and western corn rootworm are NOT Coleopteran pests, they are Lepidopteran pests

-      2.  Corn earworm are NOT Dipteran pests, they are Lepidopteran pests

-      3.  Maize billbug are NOT Dipteran pests, they are Coleopteran pests

Figure 3 (page 16)

Lines489-492

You mentioned “Frankel 1959” à provide complete citation, currently are not listed in the References section.

Need more clarification (can be added in the figure legend):

-       1.  Arrows down from secondary metabolites: does this mean that metabolites can be repellents or attractants?

-        2. The arrows down from Repellents and Attractants: what does each mean? They are not clear as each pointed out to either polyphagous insects or monophagous insects.

Check correct spelling for “repellants”: Is it repellants or repellents?

Author Response

The response to the reviewer is enclosed in the attached file

Reviewer 2 Report

Comments and Suggestions for Authors

This is an interesting review paper about an impact of metabolites in new ecologically acceptable approaches of controlling one of the most important up-to-date insect pest worldwide. The authors used the majority of the most important scientific papers from the field of study and have designed a review paper with an interesting and scientifically based structure. After carefully reading the paper I found it suitable for publication after minor revision, since I found some places in the text, which should be improved/corrected... 

Title: add "[J. E. Smith]" after "frugiperda"

Introduction

p. 2, line 62: replace "army worm" with "armyworm" and add full Latin name of "corn earworm"
p. 2, line 63: aff full Latin names of the "corn rust" and "ear rot"
p. 2, line 67: add "and some south European countries" after "... and Asia..."

Fall Armyworm - A major pest

p. 3, line 138: some references about biological control of FAW or plant pests should be mentioned in this part, for example: Trdan et al., 2020. Thirty years of research and professional work in the field of biological control (predators, parasitoids, entomopathogenic and parasitic nematodes) in Slovenia : A review. Applied science, 10, 21, art. 7468: 1-12.

Primary and secondary metabolites 

p. 6, line 277: all organisms should be written with full Latin name when they are first mentioned in the text, for example as "Oxycarenus hyalinipennis (Costa)" instead of "Oxycarenus hyalinipennis". This should be taken into account for all organisms in the paper, also Spodoptera litura (not S. litura at the time of its first mentioning, p. 6, line 279)!

I suggest to dedicate at least a short chapter to glucosinolates. This paper presents some informations in this regard... BOHINC et al. 2012. Glucosinolates in plant protection strategies: a review. Archives of biological sciences, 64, 3: 821-828.

Table 3: p. 13: replace "Spodothera litura" with "Spodoptera litura"
Figure 1: write the Latin names of harmful organisms
Figure 3: replace "repellants" with "repellents" 

Author Response

Enclosed in the attached file

Round 2

Reviewer 1 Report

Comments and Suggestions for Authors

The revised version was much improved and accepted as is.

Reviewer 2 Report

Comments and Suggestions for Authors

In its current form, the article is much better than it was in its first form. However, I suggest that the article be read by a native English speaker before final acceptance for publication. In addition, I suggest the following corrections…

Title: »J.E. Smith« should be written within brackets as [J.E. Smith]. Namely, the internal bracket must have a different shape, i.e. [ and ] from the external one, i.e. ( and )

In addition the authors should add the full Latin name of corn earworm (p. 2, line 46).

Again, the authors should check all Latin names and, at the first mention of an organism, it should be indicated with the Full Latin name.